# Potential Founder Variants in COL4A4 Identified in Bukharian Jews Linked to Autosomal Dominant and Autosomal Recessive Alport Syndrome

**DOI:** 10.3390/genes14101854

**Published:** 2023-09-23

**Authors:** Michal Levy, Lily Bazak, Noa Lev-El, Rotem Greenberg, Nesia Kropach, Lina Basel-Salmon, Idit Maya

**Affiliations:** 1The Raphael Recanati Genetic Institute, Rabin Medical Center, Petah Tikva 49100, Israelnoale1@clalit.org.il (N.L.-E.); basel@tauex.tau.ac.il (L.B.-S.); iditdanny@gmail.com (I.M.); 2School of Medicine, Tel Aviv University, Tel Aviv P.O.B 39040, Israel; 3Pediatric Genetics Unit, Schneider Children’s Medical Center, Petah Tikva 4920235, Israel; 4Felsenstein Medical Research Center, Petach Tikva 4920235, Israel

**Keywords:** COL4A4, Bukharian Jews, Alport syndrome, thin basement membrane disease, hematuria, proteinuria, Alport syndrome, COL4A4 gene, Bukharian Jewish population, Molecular diagnosis

## Abstract

Background: Alport syndrome is a hereditary disorder caused by pathogenic variants in the COL4A gene, which can be inherited in an autosomal recessive, dominant, or X-linked pattern. In the Bukharian Jewish population, no founder pathogenic variant has been reported in COL4A4. Methods: The cohort included 38 patients from 22 Bukharian Jewish families with suspected Alport syndrome who were referred the nephrogenetics clinic between 2012 and 2022. The study collected demographic, clinical, and genetic data from electronic medical records, which were used to evaluate the molecular basis of the disease using Sanger sequencing, and next-generation sequencing. Results: Molecular diagnosis was confirmed in 20/38 patients, with each patient having at least one of the three disease-causing COL4A4 variants detected: c.338G<A (p.Gly113Asp), c.3022G>A (p.Gly1008Arg), and c.871-6T>C. In addition, two patients were obligate carriers. Overall, there were 17 heterozygotes, 2 compound heterozygotes, and 3 homozygotes. Each variant was detected in more than one unrelated family. All patients had hematuria with/without proteinuria at referral, and the youngest patient with proteinuria (age 5 years) was homozygous for the c.338G>A variant. End-stage renal disease was diagnosed in two patients at the age of 38 years, a compound heterozygote for c.338G>A and c.871-6T>C. Hearing deterioration was detected in three patients, the youngest aged 40 years, all of whom were heterozygous for c.338G>A. Conclusion: This study unveils three novel disease-causing variants, c.3022G>A, c.871-6T>C, and c.338G>A, in the COL4A4 gene that are recurrent among Jews of Bukharian ancestry, and cause Alport syndrome in both dominant and recessive autosomal inheritance patterns.

## 1. Introduction

Alport syndrome is a kidney disorder manifested by recurrent or persistent hematuria. It is caused by variants in one or more of the type IV collagen genes, *COL4A3*, *COL4A4 and COL4A5*, which encode the α3, α4, α5 protein chains, respectively [1,2,3]. Alport syndrome may be inherited in different patterns—X-linked or Autosomal Dominant or Recessive. A pathogenic variant in COL4A5 confirms the diagnosis of X-linked Alport syndrome, whereas variants in COL4A3 or COL4A4 confirm the diagnosis of Autosomal Dominant or Autosomal Recessive Alport Syndrome. When there are two variants, each in a different gene (*COL4A3* and *COL4A4* and *COL4A5*), the pattern of inheritance is called digenic [4,5].

According to previous publications, the X-linked pattern accounts for 65% of cases of Alport syndrome. Symptoms are milder and occur later in women than in hemizygous men [6,7].

Autosomal recessive disease with two variants in trans within the genes-COL4A3 or COL4A4 is present in 15% of patients [8]. The natural history of autosomal recessive Alport syndrome is similar to that of X-linked disease in males. If left untreated, it progresses to end-stage renal failure in the first or second decade of life, with no difference between males and females [9]. For patients with proteinuria, starting treatment with angiotensin-converting enzyme (ACE) inhibitors early can delay the onset of end-stage renal disease [10,11].

Assessing the precise prevalence of autosomal dominant inheritance, Alport syndrome poses challenges due to potential underdiagnosis. The most common presentation typically involves hematuria, with or without proteinuria [12]. In up to 30%, according to different series, end-stage renal failure developed [13,14,15]. Based on kidney biopsy results obtained from both light and electron microscopy, Autosomal dominant Alport syndrome is occasionally known as thin membrane nephropathy (TBMN) [14,16]. According to a recent publication, 40% of patients diagnosed with TBMN by kidney biopsy had heterozygous variants in the *COL4A3* or *COL4A4* genes [14].

The Israeli population consists of many ethnic groups with distinctive genetic variations that are relatively frequent in each subpopulation [17] Ashkenazi Jews are known to carry several founder disease-causing variants for Alport syndrome: *COL4A5* c.5048G>A (NM_033380.3) p.Arg1683Gln, which causes X-linked Alport syndrome, and *COL4A3* c.40_63del (NM_000091.5) p.Leu14_Leu21del, which causes autosomal recessive Alport syndrome. Consanguineous marriages are allowed in the Jewish religion, and were common in all Jewish communities, including Ashkenazim. [18]. In 1997, Barker et al. were the first to describe the disease-causing founder variant in the *COL4A5* gene, Arg1677Gln, in Ashkenazi Jews, in three independently ascertained families. It caused a relatively mild form of nephritis with typical onset in the fourth or fifth decade [19]. In a more recent study, Webb et al. identified a homozygous *COL4A3* mutation, c.40_63del, in individuals with Alport syndrome. Extensive screening of 2017 unrelated Ashkenazi Jews yielded a carrier frequency of 1 in 183, indicating that *COL4A3* c.40_63del is a founder variant and may be an important cause of Alport syndrome in this population [20].

Deep intronic pathogenic variants, which typically evade detection through conventional sequencing methods, have been unveiled as a result of meticulous analysis of *COL4A5* mRNA extracted from urine or skin fibroblasts [21,22]. Using next-generation sequencing, Wang et al. identified four patients with negative results who were carriers of six deep intronic pathogenic variants. According to the literature and public databases (Human Gene Mutation Database and Leiden Open-Source DNA Variation Database), pathogenic splicing variants account for 14.9% to 24.5% of the *COL4A5* gene, suggesting that deep intronic *COL4A5* variants are not as rare as previously thought. As for deep intronic COL4A4 splicing variants, numerous reports have been documented to date [23,24].

The aim of this report was to describe the identification of three new disease-causing variants in *COL4A4*, suspected to be founder variants, in the Bukharian Jewish population with autosomal dominant and recessive Alport syndrome.

## 2. Methods

The cohort included all individuals of Bukharian origin with suspected Alport syndrome who were referred for counseling to a tertiary nephrogenetic clinic between 2012 and 2022. Demographic, clinical, and genetic data were collected from the medical records. The molecular basis of the syndrome was determined using linkage analysis, Sanger sequencing, and next-generation sequencing. Patients in whom clinical and genetic workup revealed a molecular diagnosis other than type IV collagen nephropathy were excluded.

The study was performed in accordance with the tenets of the Declaration of Helsinki and approved by the local institutional ethics committee.

### 2.1. Data Collection

The following data were collected from the patients’ computerized medical files: age, medical reason for referral to the nephrogenetic clinic, ethnic origin, consanguinity, dates of first and last clinic visits, presence of micro/macrohematuria and micro/macroproteinuria (defined as range of 30–299 mg/g creatinine, and **≥300 mg albumin/g** creatinine, respectively), first abnormal creatinine value (defined as above 1.2 milligrams per deciliter (mg/dL) for males and 1.0 mg/dL for females), highest creatinine value during follow-up, completion of kidney biopsy, results of electron and light microscopy and immunohistochemistry studies of kidney and skin biopsies, and presence and age at onset of hearing or visual impairment according to an ear, nose and throat specialist and ophthalmologist, respectively.

Proteinuria was determined by 24 h urine collection. Hematuria was classified into microscopic, defined as a finding of at least five red blood cells per high power field in urine sediment, and macroscopic was defined as visible blood in the urinary sediment or an appropriate anamnestic description in the medical report. Kidney failure was defined in adults according to the creatinine value and the Chronic Kidney Disease Epidemiology Collaboration equation (CKD-EPI) for glomerular filtration rate (GFR), and in children (age < 18 years), according to GFR < 60 mL/min/1.73 m^2^ for ≥3 months, irrespective of the presence or absence of kidney damage. ESRD includes patients treated by dialysis or transplantation, irrespective of the level of GFR.

### 2.2. Genetics

During the first encounter with the patient in the nephrogenetic clinic, a family history with a detailed four-generation pedigree, including both affected and unaffected individuals, was taken, and the most likely pattern of inheritance was noted.

After the genetic workup was completed, either with a negative result or detection of a disease-causing variant, a second counseling session was held to explain the results and reach decisions on extending screening to other family members, starting treatment, and counseling regarding future pregnancies, prenatal diagnosis, or preimplantation genetic diagnosis (PGD). Thereafter, patients were referred to their nephrologist for further follow-up and treatment.

Genetic testing was performed by exome sequencing or with a targeted gene panel including *COL4A5* (GenBank reference sequence, RefSeq: NM_033380.3), *COL4A3* (RefSeq: NM_000091.5), and *COL4A4* (RefSeq: NM_000092.5). In cases of an already known pathogenic COL4A4 variant, either targeted variant analysis or Sanger sequencing were performed. The resulting variants were scored on the standard five-tier system according to the guidelines of the American College of Medical Genetics and Genomics (ACMG), based on data from population- and disease-specific databases, type, and segregation of the variant [25].

The local exome database was searched for the disease-causing variants to estimate the allele frequency among individuals of Bukharian ethnicity, only when the indication for exome sequencing was not related to the clinical suspicion of Alport syndrome. Clinical information on these individuals was collected from the database to evaluate a clinical relationship to Alport syndrome.

## 3. Results

### 3.1. Study Population

During the study period, a total of 237 patients from 200 families of Jewish origin were referred to our nephrogenetic clinic for clinical evaluation and genetic analysis for suspected Alport syndrome. Of these, 38 patients from 22 families were of Bukharian Jewish origin and formed the study cohort. The mean duration of follow-up was 8.1 ± 2.1 years (Figure 1).

The primary motivation for referral to the nephrogenetic clinic varied among the 22 families as follows: obtaining a genetic diagnosis for their medical condition or family history was the primary reason in 18/22 families (81.1%), prenatal diagnosis was the main objective in 2/22 families (9%), and evaluation prior to kidney transplantation was the primary goal in 2/22 families (9%). On average, there were 1.7 affected members per family, with a range of 1 to 4. The age of patients at the time of diagnosis ranged from 5 to 59 years, with 33/38 patients (86.8%) being older than 18 years.

### 3.2. Clinical Presentation—Phenotype

At the time of referral, all patients were presented with hematuria and the majority also with proteinuria (*n* = 23, 60.5%) on 24 h urine collection, (mean 1191, range 267–3259 mg/24 h). All patients underwent audiological and ophthalmologic evaluations. Only 3/38 (7.8%) reported hearing deterioration and had an abnormal hearing test (hearing deterioration was defined by hearing thresholds of higher than 20 dB in both ears). No visual impairment was diagnosed in the cohort as per the data collection date.

Kidney biopsy studies were performed in 10/38 patients (26.3%). Electron microscopic evaluation was performed on 7/10 patients, demonstrating the pathological changes that were consistent with Alport syndrome. These changes included irregularities in the glomerular basement membrane, presence of split or fragmented basement membranes, and abnormalities in the distribution of collagen IV.

Renal failure occurred in 4/38 patients (10.5%), which progressed to end-stage disease requiring dialysis/transplantation in 2/38 (5.2%).

### 3.3. Genotype

Molecular analysis was performed in 20/38 patients (nine next-generation sequencing panel, eight Sanger sequencing for a single variant, three linkage analysis). Another two patients were obligate carriers. Two patients performed molecular analysis for Alport and had negative results. The remaining 14/38 patients in the cohort (36.8%) did not undergo genetic diagnosis. In the whole cohort, only three variants in *COL4A4* were detected (Table 1).

#### 3.3.1. Variant 1: COL4A4 c.338G>A, (NM_000092.5) p.Gly113Asp (chr2:227984645 C>T, hg19)

This sequence change replaces glycine with aspartic acid at codon 113 of the *COL4A4* protein. The glycine residue is highly conserved, and there is a moderate physicochemical difference between glycine and aspartic acid. The variant is present in population databases at a low frequency (rs766085522, gnomAD 0.005%).(Karczewski et al., 2020) (Karczewski et al., 2020) ClinVar contains an entry for this variant (Accession VCV000970268.3, Variation ID: 970268). Algorithms developed to predict the effect of missense changes on protein structure and function, support the effect of this variant on protein function (Revel deleterious 0.96, CADD 26.4). In the local database of 50 alleles of Bukharian subjects, the *COL4A4 c.338G>A*, *p.Gly113Asp* allele was identified in seven heterozygous individuals (aged 6–42 years) from five unrelated families for a carrier frequency of 1:10. Clinical information was available for four of them. Two (aged 30 and 42 years) had hematuria and two (aged 6 and 28 years) did not.

#### 3.3.2. Variant 2: COL4A4 c.3022G>A (NM_000092.5) p.Gly1008Arg (chr2:227915821 C>T, hg19)

This sequence change replaces glycine, which is neutral and non-polar, with arginine, which is basic and polar, at codon 1008 of the protein. In all patients, the variant was classified by four different clinical laboratories as a variant of uncertain significance (VUS). It is present in population databases at a low frequency (rs371172166, gnomAD0.003) [26], and was reported once in ClinVar in an individual with clinical features of autosomal dominant Alport syndrome [31] (Invitae, Variation ID: 552195). Advanced modeling of the protein sequence and biophysical properties (such as structural, functional, and spatial information, amino acid conservation, physicochemical variation, residue mobility, and thermodynamic stability) performed at Invitae indicated that this missense variant is expected to disrupt *COL4A4* protein function. The variant was not found in our local exome database.

#### 3.3.3. Variant 3: COL4A4 c.871-6T>C NM_000092.5 (chr2-227967570 A>G, hg19)

This sequence change falls in intron 14 of the *COL4A4* gene. It does not directly change the encoded amino acid sequence of the *COL4A4* protein. The variant is present in population databases at a very low frequency (rs749753913, gnomAD 0.0004%). ClinVar contains one entry for the variant (Accession VCV001002812.2, variant 1002812). Algorithms developed to predict the effect of sequence changes on RNA splicing do not predict an effect of the variant on splicing [trap score 0.041, varseak 2 (likely no splicing effect), splice AI Acceptor Loss 0.1]. The variant was not found in our local exome database.

While the available published evidence is currently insufficient to determine the role of these three variants as disease-causing, given the multiple affected patients from unrelated families in our cohort, they might be reclassified as pathogenic/likely pathogenic (Table 1).

### 3.4. Genotype-Phenotype Correlation

The clinical characteristics by genotype are presented in Table 2. (Data in the table refer only to the molecularly diagnosed patients.)

#### 3.4.1. Dominant Alport Syndrome

The cohort included 17 carriers of one of the three founder mutations in *COL4A4.* Of them, 15 were molecularly diagnosed. The average age at diagnosis was 31.3 years. All 15 patients presented with hematuria, and 9 presented with proteinuria, at an average age of 27.3 years. When present, proteinuria always followed hematuria. Renal failure developed in two patients, both heterozygous for the c.338G>A variant. None had end-stage renal disease. Three patients were found to have a hearing impairment at a mean age of 52.3 years, all heterozygous for the c.338 G>A variant.

#### 3.4.2. Recessive Alport Syndrome

There were five carriers of biallelic founder mutations of whom three were homozygous for the c.338G>A variant and two were compound heterozygous for the c.338G>A and c.871-6T>C variants. None were homozygous for the *COL4A4* c.871-6T>C nor for the c.3022G>A variant. The average age at diagnosis was 12.2 years. All five patients had hematuria at the time of referral to the nephrogenetic clinic, and all had proteinuria at an average age of 14.2 years. Two compound heterozygous carriers of the c.338G>A and c.871-6T>C variants had renal failure that progressed to end-stage renal disease at age 38 years.

## 4. Discussion

The diagnosis of Alport syndrome is suspected when an individual has the characteristic clinical features or a family history of Alport syndrome, and it is optimally confirmed by the demonstration of a disease-causing variant in the *COL4A5*, *COL4A3*, or *COL4A4* genes [32,33]. Molecular genetic testing for diagnosis in the proband and family members has advantages. First, an invasive examination for diagnosis (such as kidney biopsy) becomes unnecessary. Second, an appropriate medical follow-up can be performed for the presence of proteinuria and for hearing deterioration. Third, pre-implantation genetic testing for prevention of recessive Alport syndrome in descendants of carriers can be offered. Fourth, early detection, when the kidney has not yet been damaged, allows for clinical follow-up to detect microproteinuria, and if microproteinuria is detected, renal protective therapy can be initiated early, which postpones the need for kidney transplantation [34]. A growing body of literature suggests that timely initiation of angiotensin-converting enzyme inhibitor treatment may slow the progression of kidney disease in patients with Alport syndrome further supporting the need for early diagnosis and early nephroprotective therapy in oligosymptomatic patients [11].

We report three new possibly disease-causing variants, presumed to be founder variants, in the Bukharian Jewish population in Israel. Each variant was detected in more than one unrelated family in various inheritance patterns. When we searched for these variants in our local exome database among individuals of Bukharian ethnicity, heterozygotes for c.338G>A p.Gly113Asp were found, with a prevalence of 1:10. The other two variants were not found, probably due to a relatively small size of the database itself ~50 alleles. Each of the variants caused dominant Alport syndrome in heterozygotes and recessive Alport syndrome in homozygotes or compound heterozygotes. As expected, individuals with the autosomal recessive form of the disease were more severely affected than those with the autosomal dominant form.

Currently, more than 600 *COL4A3/COL4A4* variants have been reported. Splicing variants account for more than 10% (Human Gene Mutation Database) and are mostly located at canonical splice sites; others may be found in introns at more than 100 base pairs upstream from the exons. Daga et al. analyzed RNA extracted from urine-derived podocyte lineage cells from patients with Alport syndrome; they found that a deep intronic variant in *COL4A4* could alter splicing products [24]. The predictive tools available to us are insufficient in assessing the splicing effect of the intronic variant we have identified. Additionally, we acknowledge that the current biochemical and segregation data may not provide robust evidence for a conclusive determination. Nevertheless, our observations within the familial cohort indicate a potential genotype-phenotype correlation: individuals with compound homozygosity for the intronic variant, and the c.338 G>A variant presented with the disease at an earlier and more severe stage compared to heterozygotes for the c.338 G>A variant. To validate our findings, it is imperative to conduct functional studies at the mRNA level.

There is a plausible scenario where the variant we have identified as disease-causing might merely serve as a marker for another as-yet-undiscovered deep intronic variant. Therefore, we propose further investigations to explore the possibility of larger insertions/deletions, which may be undetectable through standard NGS methods. These additional investigations are crucial in comprehensively understanding the genetic basis of the observed phenotype.

The COL4A4 variants exhibit either dominant or recessive inheritance patterns, with the phenotype of dominant Alport syndrome being less severe in terms of a lower occurrence of kidney failure and older age of onset. For genotype–phenotype correlation in AD Alport syndrome, according to Savige et al. and Gibson et al., severe variants such as rearrangements, large deletions, truncating variants (nonsense and frameshift), associated with COL4A4 AD Alport syndrome, might primarily increase the penetrance of hematuria [35,36].

A recent study conducted by Matthaiou et al. investigated genotype–phenotype correlations in individuals with Alport syndrome. Their findings revealed that among the 74 patients with COL4A3/A4 mutations, those mutations leading to premature termination of translation were associated with a lower average age at ESRD. Specifically, in this cohort of 74 patients, 54 individuals harbored missense mutations, predominantly involving glycine substitutions, and exhibited a mean age at ESRD of 55.2 years. In contrast, the remaining 20 patients carried non-missense mutations leading to the premature termination of translation, including deletions, duplications, and splice-site mutations, and demonstrated a lower mean age of ESRD at 47.1 years. The study also reported that mutations causing a glycine substitution with bulkier amino acids displayed a correlation with a reduced age of onset of ESRD, with this correlation increasing in line with the number of side chain carbon atoms in the substituting residue [12].

In a different study, there was no difference in the age at kidney failure for truncating variants or for missense variants that were or were not Gly substitutions (*p* = 0.3); nor for Gly substitutions, splicing or missense variants (*p* = 0.90); nor Gly substitutions with Arg, Glu, or Asp or other severe non-missense variant types compared with other Gly substitutions [*p* = 0.5; (5)]. There was, however, a large intrafamilial variability in age at kidney failure [37].

The mean age of diagnosis in patients with the c.3022G>A variant is noteworthy. This variant appears to be associated with a recessive-like presentation, possibly contributing to earlier diagnosis in these individuals. Several factors may play a role in this early diagnosis, including the potential presence of another, as a yet undiscovered variant that could exacerbate the clinical presentation. While hearing impairment is commonly associated with the recessive form of Alport syndrome, our study revealed an intriguing observation. Notably, audiological impairment was observed in three patients with autosomal dominant Alport syndrome, all of whom belonged to an older age group (40–60 years old). In contrast, the autosomal recessive group primarily comprised younger individuals (5–25 years old), where hearing impairment was notably absent. This age disparity raises an important consideration. Alport syndrome’s progression and the severity of associated hearing impairment are known to vary with age. It is plausible that the younger recessive patients in our cohort have not yet reached an age where hearing impairment typically manifests. Thus, the absence of hearing issues in this group at the time of data collection may be attributed to their younger age, and the possibility of future audiological changes cannot be overlooked.

This study was limited by the small sample size and relatively short follow-up period. As Alport syndrome progressed over decades, whereas our follow-up time was restricted, we were unable to draw valid conclusions regarding long-term prognosis. In addition, our cohort did not contain any patients who were homozygous for the c.3022G>A and for the c.871-6T>C variants. Therefore, we could not ascertain the clinical presentation for a recessive form of the disorder.

The Bukharian Jewish population is characterized by an increased prevalence of several autosomal recessive diseases because of genetic drift, assortative mating, and population bottlenecks. It is possible that the variants discussed here have some recurrence in other populations, however, the frequency is likely to be much lower.

During the last decades, prenatal screening tests have been developed for recessive diseases [38]. In Israel, one of the most accepted methods for pre-pregnancy population screening is pathogenic variant-based carrier-screening panels [18]. Recessive Alport syndrome is a severe disease leading to renal failure when inherited, in both homozygotes and compound heterozygotes. Therefore, its inclusion in carrier screening might be justified. Since carriers of *COL4A4* variants resulting in dominant Alport syndrome have hematuria as a presenting symptom, individuals revealed to be carriers might benefit from improved follow-up and treatment.

In conclusion, our study highlights the recurrence of three variants—c.338G>A, c.3022G>A, and c.871-6T>C—within the *COL4A* gene among individuals of Jewish Bukharian ancestry, which can lead to Alport syndrome through both dominant and recessive inheritance patterns. Considering these findings, we recommend that, for individuals of Jewish Bukharian descent presenting with Alport syndrome or persistent hematuria, a search for these variants be prioritized as a first-tier test, before proceeding with gene panel sequencing. To validate and further elucidate the implications of these findings, essential functional studies at the mRNA level are proposed. Additionally, in parallel, deep intronic sequencing and MLPA analysis will be undertaken. These complementary investigations will enable the validation of the disease-causing potential of the intronic variant, c.871-6T>C, and offer insights into the possible presence of other deep intronic variants that may contribute to the observed phenotype. The aim is to provide a more comprehensive understanding of the genetic landscape underlying the condition, thereby advancing personalized medical care and genetic counseling for affected individuals and their families.

## Figures and Tables

**Figure 1 genes-14-01854-f001:**
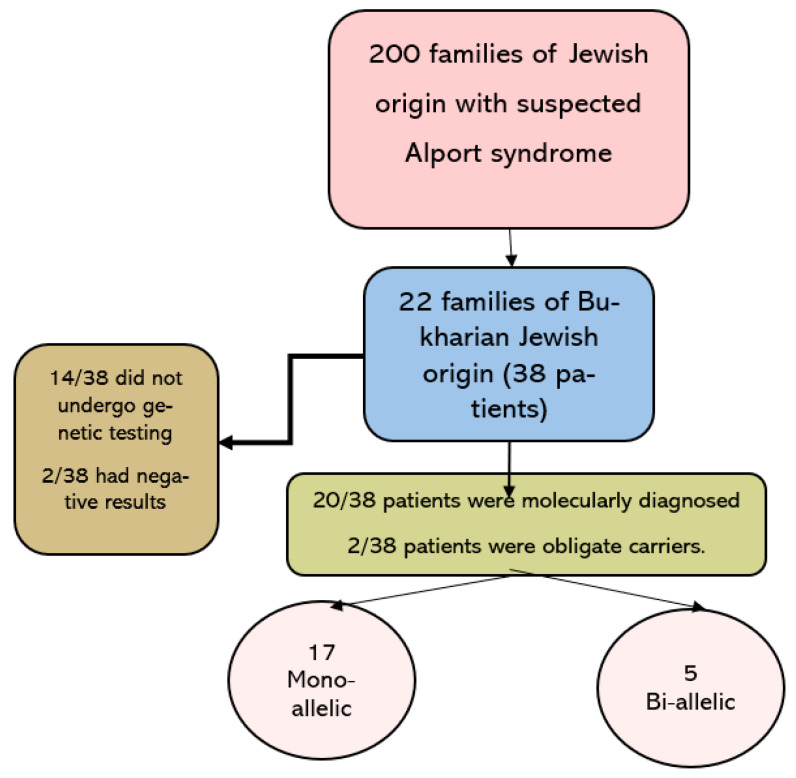
Study population.

**Table 1 genes-14-01854-t001:** Variants found in the *COL4A4* gene.

Genomic Location (GRCh37)	Variant Name	Variant Type	GnomAD [26]	Prediction Tools [27,28,29]	Previous Classification [30]	Our Classification [30]
2: 227984645	c.338G>A, NM_000092.5 p.Gly113Asp	Missense	14 het total 0.005% (Afr 9 0.04%; Lat 5 0.01%)	Revel deleterious 0.96, CADD 26.4	VUS– PM2_support, PP3	P—PS4, PM2_support, PP3, PP1_strong
2: 227915821	c.3022G>A, NM_000092.5 p.Gly1008Arg	Missense	9 het total 0.003% (Afr 6 0.2%; Lat 2 0.006%; Eur 1 0.0008%)	Revel D 0.81 CADD 39	VUS– PM2_support, PP3	P—PS4, PM2_support, PP3, PP1_strong
2: 227967570	c.871-6T>C, NM_000092.5	Intronic	1het total 0.0004% (Eur 0.0008%)	Trap score 0.041, varseak 2 * (Likely no splicing effect), splice AI Acceptor Loss 0.1	VUS–PM2_support, BP4	LP– PS4, PM2_support, PP1

VUS—variant of unknown significance; P—pathogenic; PM2—Absent from controls (or at extremely low frequency if recessive) in gnomAD, ESP, 1000 Genomes or ExAC.; PP3—Multiple lines of computational evidence support a deleterious effect on the gene or gene product (conservation, evolutionary, splicing impact, etc); PS4—Protein length changes due to in-frame deletions/insertions in a non-repeat region or stop-loss variants; PP1—Co-segregation with disease in multiple affected family members in a gene definitively known to cause the disease; BP4—Multiple lines of computational evidence suggest no impact on gene or gene product (conservation, evolutionary, splicing impact, etc). * www.varSEAK.bio (accessed on 20 August 2022), developed by JSI Medical Systems GmbH, Ettenheim, Germany.

**Table 2 genes-14-01854-t002:** Clinical characteristics per genotype *.

	AD Alport			AR Alport	
Mutation type	HET c.871-6T>C	HET c.338 G<A	HET c.3022G>A	HOMO c.338G>A	COMP. HET c.338G>A c.871-6T>C
Number pts	4	8	3	3	2
Age (year) at diagnosis, mean (range)	34.2 (26–59)	33.6 (18–54)	12.6 (5–25)	9 (7–12)	17 (16, 18)
Hematuria	4/4	8/8	3/3	3/3	2/2
Age (year) at onset, mean (range)	35.7 (26–52)	36.3 (18–65)	14.3 (7–28)	6.7 (5–8)	17 (13, 18)
Proteinuria	3/4	4/8	2/3	3/3	2/2
Age (year) at onset, mean (range)	32.3 (30–37)	30 (19–50)	15.5 (8–13)	9 (7–12)	22 (18, 26)
Renal failure	0	2/8	0	0	2/2
Age (year) at onset, mean (range)	N/A	33.5 (21–46)	N/A	N/A	37 (36, 38)
ESRD	0	0	0	0	2/2
Age (year) at onset, mean (range)	N/A	N/A	N/A	N/A	38 (38, 38)
Hearing impairment	0	3\8	0	0	0
Age (year) at onset, mean (range)	N/A	52.3 (40–60)	N/A	N/A	N/A

* Note: Data in the table refer only to the molecularly diagnosed patients. AD, autosomal dominant; AR, autosomal recessive; HET, heterozygous; HOMO, homozygous; ESRD, end-stage renal disease; N/A, not available.

## Data Availability

The datasets generated and analyzed during the current study are not publicly available due to privacy concerns. However, they are available from the corresponding author upon reasonable request.

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
