# Peer review of "Potential Founder Variants in COL4A4 Identified in Bukharian Jews Linked to Autosomal Dominant and Autosomal Recessive Alport Syndrome"

_genes, 2023, doi:10.3390/genes14101854_

Round 1
Reviewer 1 Report
The authors describe three COL4A4 gene variants which they consider as founders because they were found in more than one unrelated family. The variants were found in Bukharan Jews, a subpopulation of Jews. Although of interest in terms of founder variants which prompt easier genetic testing for presymptomatic, symptomatic and prenatal testing, I have some reservations regarding several statements and the confidence and validity of the results.
Specifically:
Line 54, Page 2: The authors state: “Approximately 20% of patients with Alport syndrome described in the medical literature had an autosomal dominant inheritance pattern”.
I would like to know how the authors derived this 20% number. Did they count all cases in the literature? In my view, if we name all heterozygous patients as autosomal dominant Alport syndrome, this percentage would be much higher.
In the same paragraph the authors comment on hearing and ocular abnormalities. I suggest they read and take into consideration a systematic review published in 2020, in Clinical Kidney J, on 777 patients.
In lines 63-76, Page 2, do the authors refer to two or three founder variants?
Line 78, Page 2: Publications in 2000 and 2012 are not considered as recent.
Lines 173-174, Page 4: How is this variant classified/evaluated to be of moderate physicochemical difference? Biochemically, one would evaluate it of major physicochemical difference as aspartic acid is a charged acidic amino acid, and much bulkier than glycine. Franklin database classifies variant p.Gly113Asp as Likely Pathogenic.
The second variant, p.Gly1008Arg, is also classified by Franklin as Likely Pathogenic.
In my opinion there is not enough evidence to consider the third variant as Likely Pathogenic or Pathogenic. The biochemical and segregation data are weak. Franklin and Clinvar classify it as VUS (towards Likely Benign) or Likely Benign, respectively.
Although the data are tempting to accept that the variant COL4A4:c.871-6T>C is pathogenic, there is not enough convincing evidence for this. Similarly, the rather severe phenotype with early age at onset of proteinuria and/or renal failure in the heterozygous patients who carry the other two mutations might be suspicious for a second variant inherited in trans that remained undetected. I suggest that the authors include in their testing method the search for larger insertions/deletions using the MLPA technique. Whole exon deletions/insertions may be missed with NGS. The authors recognise in their discussion that other true causative variants may be in linkage disequilibrium with the described variants, for which in addition to deep intronic sequencing, MLPA analysis could be revealing.
Lines 279-281, Page 7: The authors state: “In dominant Alport syndrome, by contrast, missense variants have a more severe manifestation than protein-truncating variants and large deletions, probably because of damage to the spatial structure of the protein”.
What is the evidence for this statement? I also suggest they read: Matthaiou A et al, Clin Kidney J, 2020.
No comments
Reviewer 2 Report
Ley et al. report 3 new disease-causing COL4A4 variants detected in a cohort of 38 patients from 22 from Bukharan Jewish families with suspected Alport syndrome, referred to a nephrogenetic clinic over a 10 year-period. These variants seem to be rather frequent in this population, potentially as a result of a founder effect.
General comments
The topic of the present study is of potential interest, in the moving field of the COL4A variants and the associated genotype-phenotype correlations
A number of informations (biological, histological and clinical) reported in the paper should be corrected (see below). In addition, some important information is also missing (see below too).
Overall the paper definitely requires a major revision, to clarify a long listing of points, correct some inaccuracies and provide additional information.
The authors include in the current report 14 patients who “ did not undergo a genetic diagnosis”. Why are they included in the present study?
Specific comments
1.In the introduction, line 45 "almost all women with X linked disease are heterozygous", why do the authors say “almost all”? Please adapt or remove the sentence.
2. Line 79, reference 22 does not refer to mRNA from urine or skin, please correct.
3. In reference 21, Wang et al only studied COL4A5 (and not COL4A3) in the skin; so please correct or add a reference for deep intronic variants in COL4A3 gene. The authors mention that no deep intronic variant have been described so far in COL4A4 and then in the discussion they mention the study of Daga et al. Please clarify.
4. In the Data collection section, proteinuria is not defined (values? Cut off?), kidney failure is also not defined and classified (level of eGFR? KDIGO classification?) neither is end-stage renal disease. Creatinine values alone do not automatically lead to a GFR estimate in children, do the authors use a specific formula (Schwartz? Other?)
5.In the results and study population section, what do the authors mean by "the reason for referral was the significance of a genetic diagnosis"?
6. In the Clinical Presentation section
- proteinuria was not defined and the amount of proteinuria is not mentioned.
- The authors mention “hearing deterioration”, what do they mean? Hearing aid? Other? Did all the patients have an audiological and ophthalmologic evaluation? If not, the data are only retrospective and rather useless.
- Moreover, 3 patients with autosomal dominant Alport syndrome have audiological impairment and none in the autosomal recessive group, this is quite unusual, can the authors comment on that? No data about ophthalmologic findings?
- Regarding kidney biopsies, they mention TBMD seen by light microscopy and immunofluorescence but the definition on this type of nephropathy relies only on electronic microscopy, please correct
. What do the authors mean by "pathological changes consistent with Alport syndrome" on immunofluorescence and EM line 157?
7. Definition of renal failure line 158?
8. Table 1 : please explain the abbreviations for non-geneticists (PS4,PM2) etc...
Table 2 : can the authors comment on the age of diagnosis (quite young 12.6 years old) in patients with c.3022G? Same comment for the amount of proteinuria and the definition of renal failure
9. How do the authors explain that variant 2 and 3 were not found in the local data base from Bukharan subjects?
10. In the discussion, please refer to the recent classification of Kashtan et al (Kidney international 2018) and J Savige (cJASN 2022) to make the diagnosis of Alport syndrome (rather than to references 26-27-28).
11. There is a mistake in line 270,this reviewer suspects the authors mean COL4A3 and A4 (rather than A5)?
12. Reference 10 and 11 are almost identical and 30 identical to 10, the exact reference for the EARLYPROTECT trial is ref 11.
13. The authors mention in the discussion (line 289) the duration of follow-up(8.1±2.1 years). This should be in the results section .
Minor comments
Is the correct spelling “Bukharan”(title) or “Bukharian” (line 87)?
In the abstract, the authors say that their study “confirms” that 3 disease-causing variants”…… As the 3 variants are novel, this wording is perhaps a bit shy?
see above for the detailed comments
Reviewer 3 Report
1. What is the main question addressed by the research?
The main question raised by the authors is the identification of 3 new disease-causing 86 variants in COL4A4, suspected to be founder variants, in the Bukharian Jewish population 87 with autosomal dominant and recessive Alport syndrome, The authors of this publication have attempted to answer this question.
2. Do you consider the topic original or relevant in the field? Does it address a specific gap in the field?
I believe that the publication is original, but it concerns a relatively small ethnic group. No significant dissemination of results to the remaining population
3. What does it add to the subject area compared with other published material?
The publication is a summary of genetic research relating to the causes of a rare disease, for which faster detection is more likely to slow down clinical progress
4. What specific improvements should the authors consider regarding the methodology? What further controls should be considered?
I believe that the methodological part of genetic research should be described in more detail. However, the results are presented correctly. Possibly, there is a lack of a graphic representation of the developed results
5. Are the conclusions consistent with the evidence and arguments presented and do they address the main question posed?
6. The conclusions are summarized appropriately and answer to the main thesis of the work
Are the references appropriate?
The references are relevant to the topic at hand, covering both older and recent research
7. Please include any additional comments on the tables and figures.
The publication does not contain figures and the presented tables are correctly constructed. I have no additional comments on this part of the work
Author Response
Response
Thank you for your valuable suggestions. As per your suggestion, we have elaborated on the methodological part of genetic research.
Round 2
Reviewer 2 Report
the paper is definitely much improved. At line 193, delete "Genotype phenotype correlation in AD Alport syndrome" (probably a remnant of a previous version). This can be dealt with at the proofs stage
no specific comments